# Continuous Piperacillin-Tazobactam Infusion Improves Clinical Outcomes in Critically Ill Patients with Sepsis: A Retrospective, Single-Centre Study

**DOI:** 10.3390/antibiotics11111508

**Published:** 2022-10-29

**Authors:** Dong-gon Hyun, Jarim Seo, Su Yeon Lee, Jee Hwan Ahn, Sang-Bum Hong, Chae-Man Lim, Younsuck Koh, Jin Won Huh

**Affiliations:** 1Department of Pulmonary and Critical Care Medicine, Asan Medical Centre, University of Ulsan College of Medicine, Seoul 05505, Korea; 2Department of Pharmacy, Asan Medical Centre,University of Ulsan College of Medicine, Seoul 05505, Korea

**Keywords:** piperacillin-tazobactam, intensive care unit, critical care, pneumonia, intravenous infusion

## Abstract

Continuous infusion of beta-lactam antibiotics has emerged as an alternative for the treatment of sepsis because of the favourable pharmacokinetics of continuous infusion. This study aimed to evaluate the survival benefits of continuous vs. intermittent infusion of piperacillin-tazobactam in critically ill patients with sepsis. We retrospectively conducted a single-centre study of continuous infusion vs. intermittent infusion of piperacillin-tazobactam for adult patients who met the Sepsis-3 criteria and were treated at a medical ICU within 48 h after hospitalisation between 1 May 2018 and 30 April 2020. The primary outcome was mortality at 28 days. A total of 157 patients (47 in the continuous group and 110 in the intermittent group) met the inclusion criteria for evaluation. The 28-day mortality rates were 12.8% in the continuous group and 27.3% in the intermittent group (*p* = 0.07). However, after adjustment for potential covariables, patients in the continuous group (12.8%) showed significantly lower mortality at 28 days than those in the intermittent group (27.3%; adjusted hazard ratio (HR), 0.31; 95% confidence interval (CI), 0.13–0.79; *p* = 0.013). In sepsis patients, continuous infusion of piperacillin-tazobactam may confer a benefit regarding the avoidance of mortality at 28 days compared with intermittent infusion.

## 1. Introduction

Sepsis persists as a major cause of mortality worldwide, causing millions of deaths annually [1]. Beta-lactam antibiotics with a relatively wide antimicrobial range are one of the main types of antibiotic initially administered to patients with sepsis [2]. The standard practice for infusing beta-lactam antibiotics is intermittent administration as a bolus or short infusion [3]. However, beta-lactam antibiotics show antibacterial activity depending on the time during which their concentration is maintained above the minimum inhibitory concentration (MIC) of the causative bacteria [4,5,6,7]. Based on this characteristic of beta-lactam antibiotics, continuous infusion of beta-lactam has emerged as an alternative to intermittent infusion [8]. Several studies of continuous beta-lactam antibiotic infusion have investigated the therapeutic effect on sepsis, but previous studies with heterogeneous populations or various types of beta-lactam antibiotics reported conflicting outcomes [9,10,11,12].

Piperacillin-tazobactam is a beta-lactam antibiotic commonly prescribed for the treatment of sepsis [13]. Recently, three meta-analyses including piperacillin-tazobactam demonstrated clinical benefits favouring continuous infusion of piperacillin-tazobactam over its intermittent infusion in patients with sepsis [14,15,16]. However, the wider application of continuous antibiotic infusion still remains limited in actual clinical practice because of a number of important factors, such as stability, drug–drug incompatibility of antibiotics, or, concerns about the emergence of antimicrobial resistance [17,18,19]. Therefore, the aim of this study was to evaluate whether continuous infusion of piperacillin-tazobactam leads to improved clinical outcomes of critically ill patients with sepsis in a real-world intensive care unit (ICU) setting.

## 2. Results and Discussion

We screened 652 eligible patients for inclusion during the study period, of whom 491 were excluded for different reasons: 202 who received piperacillin-tazobactam for less than 48 h in the medical ICU, 287 who did not start continuous antibiotic infusion within 48 h after hospitalisation, and 2 who concurrently used other beta-lactam antibiotics with piperacillin-tazobactam (Figure 1). Of the remaining 161 patients, 157 met the Sepsis-3 criteria. A total of 157 patients were thus included for analysis: 47 patients (29.9%) in the continuous group and 110 (70.1%) in the intermittent group. No issues of drug stability or drug–drug compatibility were reported during the study period.

### 2.1. Baseline Demographics and Clinical Characteristics

Baseline demographics, including age, sex, body mass index, and pre-existing conditions, were similar between the two groups (Table 1). No significant differences were found between the continuous and intermittent groups in terms of organ dysfunction at baseline, severity of illness, and primary site of infection. Patients received piperacillin-tazobactam treatment for medians of 7.3 days (IQR, 4.0–10.0) and 8.0 days (IQR, 5.5–11.4) in the continuous and intermittent groups, respectively (*p* = 0.137). Most patients were treated with concomitant agents including teicoplanin (26.1%), levofloxacin (70.1%), and trimethoprim-sulfamethoxazole (14.0%). Laboratory variables (lactate, C-reactive protein [CRP], and white blood cell [WBC] count) associated with infection were similar between the two groups. There was a significant difference of the median hospital stay before inclusion (1.0 day [IQR, 0.0–1.0] in the continuous group vs. 0.0 days [IQR, 0.0–0.0] in the intermittent group, *p* = 0.018). However, similar numbers of patients in the two groups received interventions for organ support, including mechanical ventilation.

### 2.2. Microbiological Data

A total of 71 (45.2%) of the 157 patients had a pathogen confirmed by culture. At least one causative organism was identified in 15 patients (9.6%) in blood culture, 59 patients (37.6%) in sputum culture and 6 patients (3.8%) in other culture. Nine patients (5.7%) had infections by multiple microbes. The two groups had similar microbiological profiles at baseline (Table 2). The isolates from blood were mainly Gram-positive bacteria in both groups (100% in the continuous group vs. 70.0% in the intermittent group, *p* = 0.505). The organisms predominantly isolated from sputum culture in both groups were Gram-negative bacteria (46.7% vs. 43.2%, *p* = 0.814). Among patients with the identified pathogens, the pathogens of 27 patients (17.2%) were susceptible to piperacillin-tazobactam. There were no differences in the susceptibility to piperacillin-tazobactam from three cultures between the two groups.

### 2.3. Primary Outcome

The 28-day mortality rates were 12.8% in the continuous group and 27.3% in the intermittent group (*p* = 0.07) (Figure 2). We further adjusted the association between type of infusion and mortality at 28 days, using confounders including congestive heart failure, cardiovascular dysfunction, renal replacement therapy, positive blood culture, Sequential Organ Failure Assessment (SOFA) score, primary site of infection, lactate level, and number of antibiotics at pre-enrolment (Appendix A). After adjustment for potential covariables, patients who received continuous infusion of piperacillin-tazobactam had a 69% lower likelihood of mortality at 28 days than those who received intermittent infusion (adjusted HR, 0.31; 95% CI, 0.12–0.78; *p* = 0.013).

### 2.4. Secondary Outcomes

After adjusting for confounders such as organ dysfunction, site of infection, SOFA score, APACHE II score, septic shock, organ support, lactate level, CRP level, other antibiotics, and positive blood culture (Appendix A), the continuous group had a higher probability of being free from invasive ventilation at 14 days than the intermittent group (HR, 1.77; 95% CI, 1.10–2.84; *p* = 0.018), but they had similar rates at 28 days (HR, 1.25; 95% CI, 0.76–2.06; *p* = 0.376) (Figure 3). Similarly, the rate of discharge alive from the ICU at 14 days was significantly higher in the continuous group than in the intermittent group (HR, 1.95; 95% CI, 1.27–3.01; *p* = 0.002), but there was no difference in this variable at 28 days (HR 1.21; 95% CI, 0.81–1.82; *p* = 0.347) (Appendix A).

The rate of microbiological response did not differ significantly between the two groups (*p* = 0.538): 13 patients (27.7%) in the continuous group and 24 patients in the intermittent group (21.8%) showed such a response (Table 3). The rates of normalisation of WBC on day 7 were similar in the two groups (20.5% vs. 24.3%, *p* = 0.612). However, patients in the continuous group had higher rates of both normalisation of CRP (9.3% vs. 1.9%, *p* = 0.030) and lactate clearance (81.0% vs. 51.0%, *p* = 0.009) on day 7 than those in the intermittent group. In addition, there was a high tendency for normalisation of procalcitonin on day 7 in the continuous group (19.0%) compared with the intermittent group (10.0%, *p* = 0.072). There were no significant differences between the two groups with respect to length of hospital stay (21.0 days [IQR, 13.0–44.0] in the continuous group and 19.0 days [IQR, 11.0–34.5] in the intermittent group, *p* = 0.476) or with respect to length of ICU stay (7.0 days [IQR, 4.0–10.0] in the continuous group and 8.0 days [IQR, 5.0–15.3] in the intermittent group, *p* = 0.116).

## 3. Discussion

In this retrospective, single-centre study, we observed that the continuous infusion of piperacillin-tazobactam for treating sepsis in adult ICU patients who met the Sepsis-3 criteria resulted in a lower mortality rate at 28 days than in those with intermittent infusion. Furthermore, continuous infusion was associated with freedom from ventilation and discharge from the ICU at 14 days. There were significant differences between the continuous group and the intermittent group with respect to the normalisation of CRP and lactate clearance on day 7. This shows that continuous infusion may be more effective than intermittent infusion during the early inflammatory phase in critically ill adult patients with sepsis.

Conflicting findings have been reported about the efficacy of continuous infusion of beta-lactam antibiotics including piperacillin-tazobactam [20,21,22]. However, a meta-analysis of prospective clinical studies for severe sepsis demonstrated the superiority of continuous infusion over intermittent infusion in terms of patient survival [14]. A particularly noteworthy feature of this study is the finding in subgroup analysis that piperacillin-tazobactam was significantly associated with improved hospital mortality at 30 days. Another meta-analysis including 1876 adult patients comparing prolonged vs. short-term intravenous infusion of antipseudomonal beta-lactams for the treatment of sepsis also showed lower mortality in patients receiving prolonged infusions [15]. In a recent meta-analysis of 13 studies, the subgroup analysis of studies published in or after 2015 reported a significant improvement in hospital mortality or clinical cure in the prolonged infusion group [16]. Our observation that mortality at 28 days was lower with continuous infusion than with intermittent infusion of beta-lactam antibiotics among sepsis patients is consistent with the findings of these meta-analysis studies. These observations are also consistent with a recent retrospective, paediatric cohort study in which a subgroup analysis showed that critical care patients who received continuous infusion had significantly lower all-cause mortality within 30 days than those who did not [23]. Given these findings, the administration of piperacillin-tazobactam by continuous infusion in critically ill patients with sepsis may be associated with decreased hospital mortality compared with intermittent infusion.

Critically ill patients exhibit pathophysiological changes that might lead to markedly different antibiotic concentrations compared with those of other hospitalised patients [24]. The continuous infusion of beta-lactam antibiotics has been shown to achieve appropriate pharmacokinetic targets, even in the presence of ICU-associated pathophysiological disturbances [21,22]. We found that the administration of piperacillin-tazobactam by continuous infusion is associated with a higher rate of normalisation of CRP and lactate clearance on day 7 than intermittent infusion. Previous studies showed that changes in CRP concentration were associated with mortality and response to antibiotics of critically ill patients [25,26]. In addition, lactate clearance was an effective tool for predicting the prognosis of sepsis patients [27]. Considering the improvements of CRP and lactic acid, it is reasonable to interpret that continuous infusion is advantageous for resolving infection by obtaining an appropriate concentration of antibiotics for sepsis patients with pharmacokinetic variations [25,26,27,28]. Although the continuous infusion of piperacillin-tazobactam was not associated with improvement of bacteriological efficacy documented by a higher microbiological response, this result might be associated with the low rate of positive cultures. A large number of patients received empirical treatment without microbiological confirmation in this study. The causative pathogen was not identified in about half of the critically ill patients. Approximately 90% of patients were diagnosed with pneumonia and only in 37.6% of patients was the sputum culture positive in this study. In general, the abundance of pathogens in critically ill patients may often be underestimated [29]. In particular, patients diagnosed with pneumonia were reported to have lower sensitivity of pathogen detection [30]. Similar to our results, other studies found no significant differences in bacteriological efficacy between the continuous infusion and intermittent infusion of piperacillin-tazobactam [11,21,23].

This study has several limitations that should be taken into account when interpreting the results. First, this was a retrospective study at a single-centre. Bias could clearly have had an influence on this study. Patients were only recruited from one centre in one country, which may have limited the applicability of the results to other institutions and locations. Second, selection bias may also have occurred in this study because the decision about the infusion method to be used was performed by an intensivist. Third, most patients simultaneously received concomitant antibiotics, such as levofloxacin or teicoplanin, with piperacillin-tazobactam. These antibiotics may have affected the benefit on outcomes. However, the interference of these agents with outcomes seemed to be insignificant because there was no difference in the proportion of other antibiotic use between the two groups. In addition, the use of other antibiotics was not significantly associated with mortality at 28 days or discharge alive from the ICU at 14 and 28 days in the multivariable analysis. Although there was an association between other antibiotic use and ventilator weaning, patients with continuous rather than intermittent infusion had a significantly higher likelihood of ventilator weaning at 14 days, even after adjustment for the use of other antibiotics. Fourth, the application of continuous infusion in actual practice was insufficient; indeed, only 29.9% of 157 patients received continuous infusion during the study period. Finally, we did not monitor the pharmacokinetics of piperacillin-tazobactam, which is an important variable correlating with the outcome. The lack of pharmacokinetic data limits our ability to interpret our results. Nevertheless, the strengths of this study include the homogeneous cohort of patients with sepsis treated with piperacillin-tazobactam only, and the focus on mortality as the primary outcome, allowing relatively robust assessment of the improvement in survival.

## 4. Materials and Methods

### 4.1. Study Design

We conducted a retrospective analysis to compare clinical outcomes between sepsis patients who received continuous infusion and intermittent infusion of piperacillin-tazobactam at two medical ICUs of Asan Medical Centre between 1 May 2018 and 30 April 2020. The intermittent infusion of piperacillin-tazobactam was traditionally the routine approach in our institution. Beginning 1 May 2018, the continuous infusion of piperacillin-tazobactam, was introduced at these two participating medical ICUs. Guidance and education of pharmacy, nursing and medical staff was completed prior to the implementation of the protocol. The application of the continuous infusion protocol was finally determined at the treating medical staff’s discretion, considering the practicalities such as the availability of equipment and the presence of an accessible central venous line. If a causative organism resistant to piperacillin-tazobactam was subsequently identified, the antibiotic was changed to another one appropriate for the organism. In the case of patients for whom the pathogen was unknown, empirical piperacillin-tazobactam was maintained unless the clinical status deteriorated. Patients in both groups were treated during their ICU stay by an intensivist with standard intensive care.

### 4.2. Selection of Patients

All patients who were admitted to the medical ICUs during the observation period were screened for eligibility. Patients were included if they met all of the following criteria: (1) adult patients (age ≥ 18 years) who were diagnosed with sepsis according to the Sepsis-3 definitions in the previous 48 h; (2) the infusion of piperacillin-tazobactam was initiated at the ICU within 48 h after hospitalisation; and (3) patients received piperacillin-tazobactam for at least 48 h during their ICU stay. Patients who did not start to receive piperacillin-tazobactam within 48 h after hospitalisation, or who received other beta-lactam antibiotics simultaneously were excluded.

### 4.3. Infusion Protocol of Piperacillin-Tazobactam

Patients received either continuous infusion (continuous group) or intermittent intravenous (intermittent group) of piperacillin-tazobactam. The patients in the continuous group received a loading dose of 4.5 g of piperacillin-tazobactam in 50 mL of normal saline infused by an intravenous line over 30 min, followed immediately by the continuous infusion of 18 g of piperacillin-tazobactam in 240 mL of saline over 24 h. The patients in the intermittent group received 4.5 g of piperacillin-tazobactam in 50 mL of normal saline infused by an intravenous line over 30 min every 6 h. Piperacillin/tazobactam dosing regimens were adjusted for creatinine clearance. Information about drug–drug compatibility was provided to medical ICU staff, along with a continuous infusion protocol.

### 4.4. Outcomes and Measurements

The primary outcome was mortality at 28 days. Secondary outcomes included discharge alive from the ICU at days 14 and 28; freedom from an invasive ventilator at days 14 and 28; microbiological response; normalisation of WBC, CRP and procalcitonin on day 7; and more than 50% clearance of lactate on day 7. Lengths of stay in the ICU and hospital were also evaluated. Freedom from mechanical ventilation and discharge alive from the ICU were composite outcomes combining mortality with ventilator time or ICU stay. The main events in these outcomes were ventilator weaning or ICU discharge within 14 or 28 days, while death within 28 days was considered a competing risk event with zero ventilation or ICU days. Microbiological response was defined as eradication of the pathogen upon the completion of treatment among patients with positive culture. We collected data for WBC count, CRP, and procalcitonin as inflammatory markers, and the change of lactate level as a marker of metabolic status. WBC count, CRP, procalcitonin and lactate levels were recorded from the start day of piperacillin-tazobactam to days 3, 5, and 7. Among patients who had elevated laboratory data, both the presence and the duration of resolution (normalisation or more than 50% clearance) were examined.

We used data collected electronically and extracted from electronic medical records. These data included information on demographics and variables related to microbiological and clinical outcomes. To assess disease severity for patients with sepsis, SOFA and APACHE II scores were calculated and recorded within 24 h of inclusion (Appendix A). The baseline SOFA score was assumed to be zero in patients without known pre-existing organ dysfunction. The focus of infection, duration of piperacillin-tazobactam, cause of piperacillin-tazobactam termination, concomitant antibiotic use, laboratory variables, organ support at baseline and pre-enrolment status were also recorded. A causative pathogen was identified by microbiological cultures taken from the suspected infection site, including blood culture in all patients, immediately before commencement of the antibiotic. The susceptibility to piperacillin-tazobactam was reviewed.

### 4.5. Statistical Analysis

Data are presented as numbers with percentages for categorical variables and means plus standard deviations or medians with interquartile ranges (IQR) for continuous variables. Chi-squared test or Fisher’s exact test was used to compare categorical variables, while Student’s *t*-test or Mann–Whitney *U*-test was used for comparing continuous variables with a normal or non-normal distribution. For time-to-event analysis, the Kaplan–Meier method was used to estimate survival curves, whereas a log-rank test was used to test the significance of the differences. Furthermore, cumulative incidence curves were generated for both the time to discharge alive from the ICU and the time to freedom from invasive ventilation. The primary outcome of mortality at 28 days was adjusted by multivariable modelling of the association with the type of piperacillin-tazobactam administration using Cox proportional hazards regression analysis. Time-to-event analysis was right-censored at 28 days. Adjusted models were fitted by clinically likely confounders, considering the problem of collinearity. We selected covariables with statistical differences in comparison between groups or *p*-values < 0.10 in the univariable analysis. The results are presented as hazard ratio (HR) with 95% confidence interval (CI). Adjustments for discharge alive from the ICU and being free from a ventilator as secondary outcomes were also performed by the Cox proportional hazards regression model with covariables. If a subject died within the right-censored day after inclusion, days of discharge alive from the ICU and freedom from a ventilator were calculated as zero. The proportional hazard assumption was assessed through the inspection of Schoenfeld residuals. Two-sided *p* values < 0.05 were considered to indicate significance. All analyses were performed using SPSS version 26.0 (IBM Corporation, Armonk, NY, USA) software and R software version 4.1.2 (R Core Team).

## 5. Conclusions

This retrospective, single-centre study showed that the continuous infusion of piperacillin-tazobactam was associated with low mortality at 28 days among critically ill patients with sepsis in the ICU. Our findings suggest that continuous infusion of empirical piperacillin-tazobactam for treating sepsis, especially caused by severe pneumonia, may be helpful to improve clinical outcomes. These findings should be further explored in multi-centre, blinded, prospective, randomised studies.

## Figures and Tables

**Figure 1 antibiotics-11-01508-f001:**
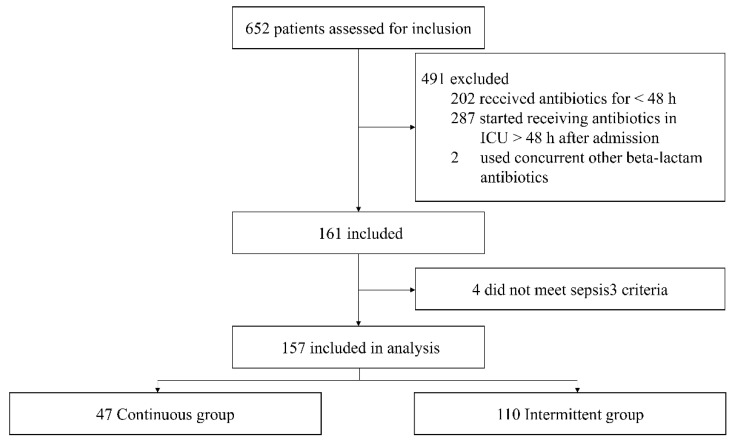
Flowchart of all excluded and included patients. ICU, intensive care unit.

**Figure 2 antibiotics-11-01508-f002:**
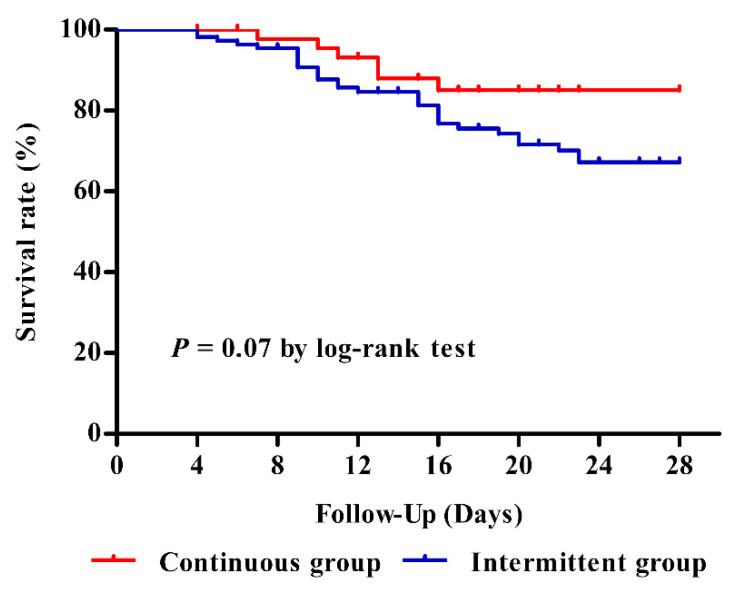
Kaplan–Meier curves of mortality at 28 days in patients with sepsis according to the type of administration of piperacillin-tazobactam.

**Figure 3 antibiotics-11-01508-f003:**
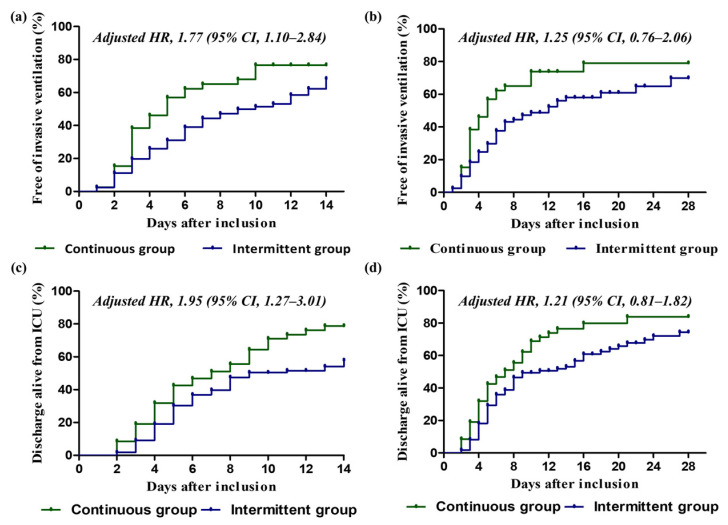
Cumulative incidence functions for patients in the continuous and intermittent groups. (**a**) Rate of freedom from invasive ventilation at 14 days. (**b**) Rate of freedom from invasive ventilation at 28 days. (**c**) Rate of discharge alive from ICU at 14 days. (**d**) Rate of discharge alive from ICU at 28 days. Continuous group (green) and intermittent group (blue) are displayed, and hazard ratio and 95% confidence interval are provided, of which a ratio of more than 1.00 indicates a higher probability of discharge alive from ICU or freedom from invasive ventilation in the continuous group than in the intermittent group. For each panel, adjusted models were performed by Cox hazard regression analysis. Co-variable selection was based on statistical significance and/or associations in the scientific literature. HR, hazard ratio; CI, confidence interval; ICU, intensive care unit.

**Table 1 antibiotics-11-01508-t001:** Baseline demographics and clinical characteristics of patients by the type of administration of piperacillin-tazobactam.

Variable	Continuous (*n* = 47)	Intermittent (*n* = 110)	*p*-Value
Age, yr	68.0 (62.0–78.0)	69.0 (60.0–75.3)	0.539
Sex, male	30 (63.8)	79 (71.8)	0.320
BMI, kg/m^2^	22.5 ± 4.2	22.7 ± 4.0	0.831
Co-existing condition			
Congestive heart failure	2 (4.3)	10 (9.1)	0.512
Respiratory failure	12 (25.5)	15 (13.6)	0.070
Chronic liver disease	1 (2.1)	8 (7.3)	0.281
End-stage renal disease	3 (6.4)	15 (13.6)	0.191
Immunocompromised	7 (14.9)	15 (13.6)	0.835
Organ dysfunction			
Cardiovascular	30 (63.8)	68 (61.8)	0.812
Respiratory	45 (95.7)	106(96.4)	1.000
Renal	16 (34.0)	41 (37.3)	0.700
Haematological	15 (31.9)	53 (48.2)	0.060
Severity of illness			
SOFA score	9.0 (6.0–10.0)	9.0 (6.8–12.0)	0.337
APACHE II score	17.2 ± 5.4	18.4 ± 6.5	0.290
Septic shock	15 (31.9)	37 (33.6)	0.834
Primary site of infection *			
Pulmonary	43 (91.5)	97 (88.2)	0.541
Abdominal	1 (2.1)	2 (1.8)	1.000
Urinary	1 (2.1)	1 (0.9)	0.510
Skin or soft tissue	1 (2.1)	0 (0.0)	0.299
Blood	1 (2.1)	5 (4.5)	0.670
Central nervous system	0 (0.0)	1 (0.9)	1.000
Others **	0 (0.0)	3 (2.7)	0.555
Unknown	2 (4.3)	8 (7.3)	0.724
Parameters of piperacillin-tazobactam			
Duration of antibiotic, days	7.3 (4.0–10.0)	8.0 (5.5–11.4)	0.137
Cause of termination, *n* (%)			0.106
Change into other antibiotics	20 (42.6)	56 (50.9)	
Cessation of antibiotic use	26 (55.3)	44 (40.0)	
Death	1 (2.10)	10 (9.1)	
Concomitant antibiotic			
Teicoplanin	14 (29.8)	27 (24.5)	0.493
Levofloxacin	34 (72.3)	76 (69.1)	0.684
TMP-SMX	8 (17.0)	14 (12.7)	0.478
Other antibiotics ***	6 (12.8)	13 (11.8)	0.868
Laboratory variables			
Lactate level, mmol/L	1.6 (1.2–2.8)	1.9 (1.3–2.5)	0.965
C-reactive protein, mg/dL	9.5 (3.8–17.5)	10.4 (3.8–19.8)	0.697
White blood cell count, × 10^3^/L	11.9 (8.8–17.3)	11.2 (5.7–15.6)	0.125
Pre-enrolment status			
ICU LOS, days	0.0 (0.0–1.0)	0.0 (0.0–0.0)	0.002
Hospital LOS, days	1.0 (0.0–1.0)	0.0 (0.0–1.0)	0.018
Number of antibiotics	1.0 (0.0–2.0)	0.0 (0.0–1.0)	<0.001
Organ support at baseline			
Mechanical ventilation	37 (78.7)	81 (73.6)	0.499
RRT	5 (10.6)	20 (18.2)	0.237
ECMO	3 (6.4)	3 (2.7)	0.365

Abbreviations: BMI, body mass index; SOFA, Sequential Organ Failure Assessment score; APACHE, Acute Physiology and Chronic Health Evaluation; TMP-SMX, trimethoprim sulfamethoxazole; ICU, intensive care unit; LOS, length of stay; RRT, renal replacement therapy; ECMO, extracorporeal membrane oxygenation. Data are reported as *n* (%), mean (standard deviation), or median (interquartile range). * Multiple sites of primary infection were identified in nine patients. ** Others of infection consisted of mediastinitis, pericarditis, and vertebral osteomyelitis. *** Other antibiotics included vancomycin, metronidazole, ciprofloxacin, moxifloxacin, amikacin, colistin, isoniazid, rifampicin, and ethambutol.

**Table 2 antibiotics-11-01508-t002:** Microbiological profile of patients by the type of administration of piperacillin-tazobactam.

Microbiological Profile *	Continuous (*n* = 47)	Intermittent (*n* = 110)	*p*-Value
Positive blood culture (*n* = 15)	5 (10.6)	10 (9.1)	0.771
Gram-positive	5 (100.0)	7 (70.0)	0.505
Gram-negative	1 (20.0)	4 (40.0)	0.600
MRSA	0 (0.0)	3 (30.0)	0.505
Susceptible to piperacillin-tazobactam	1 (20.0)	4 (40.0)	0.600
Positive sputum culture (*n* = 59)	15 (31.9)	44 (40.0)	0.338
Gram-positive	3 (20.0)	9 (20.5)	1.000
Gram-negative	7 (46.7)	19 (43.2)	0.814
MRSA	0 (0.0)	2 (4.5)	1.000
Multidrug-resistant GNB	2 (13.3)	4 (9.1)	0.638
*Pneumocystis jirovecii*	3 (20.0)	7 (15.9)	0.704
Others **	3 (20.0)	17 (38.6)	0.188
Susceptible to piperacillin-tazobactam	5 (33.3)	14 (31.8)	1.000
Other culture (*n* = 6)	2 (4.3)	4 (3.6)	1.000
Gram-positive	0 (0.0)	2 (50.0)	0.467
Gram-negative	2 (100.0)	2 (50.0)	0.467
Multidrug-resistant GNB	1 (50.0)	0 (0.0)	0.333
Susceptible to piperacillin-tazobactam	1 (50.0)	3 (75.0)	1.000

Abbreviations: MRSA, Methicillin-resistant *Staphylococcus*
*aureus*; GNB, Gram-negative bacilli. Data are reported as *n* (%). * Nine patients had multiple pathogens identified on two other cultures. ** Others included mycobacterium, fungus, and virus.

**Table 3 antibiotics-11-01508-t003:** Clinical outcomes according to the type of infusion of piperacillin/tazobactam.

Outcome Measure	Continuous (*n* = 47)	Intermittent (*n* = 110)	*p*-Value
Microbiological response	13 (27.7)	24 (21.8)	0.538
Clinical improvement, n (%)			
Normalisation of WBC on day 7 (*n* = 97)	9 (28.1)	26 (40.0)	0.272
Normalisation of CRP on day 7 (*n* = 151)	4 (9.3)	2 (1.9)	0.030
Normalisation of procalcitonin on day 7 (*n* = 81)	4 (19.0)	6 (10.0)	0.072
Lactate clearance (≥50%) on day 7 (*n* = 72)	17 (81.0)	26 (51.0)	0.009
LOS, median (IQR) *			
Hospital days	21.0 (13.0–44.0)	19.0 (11.0–34.5)	0.476
ICU days	7.0 (4.0–10.0)	8.0 (5.0–15.3)	0.116

ICU, intensive care unit; WBC, white blood cell; CRP, C-reactive protein; IQR, interquartile range; LOS, length of stay. Data are reported as *n* (%) or median (IQR). * Duration was calculated from the initiation of piperacillin-tazobactam.

## Data Availability

Not applicable.

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
