# Peer review of "Continuous Piperacillin-Tazobactam Infusion Improves Clinical Outcomes in Critically Ill Patients with Sepsis: A Retrospective, Single-Centre Study"

_antibiotics, 2022, doi:10.3390/antibiotics11111508_

Round 1

Reviewer 1 Report

Thank you for the opportunity to review this very interesting manuscript: “Continuous piperacillin-tazobactam infusion improves clinical outcomes in critically ill patients with sepsis: a retrospective, single center study”.

Overall the manuscript is very well written, the methodology is strong, the results adequately presented and discussion adequate.

In introduction page 1-2, lines 44-47, the authors refer as a limitation to clinical implementation of continuous infusion of piperacillin-tazobactam concerns about drug stability, drug-drug incompatibility of antibiotics and concerns about the emergence of antimicrobial resistance. These are very important issues and the authors should return to this subject in the results (if they have any, namely practical problems concerning drug-drug incompatibility) or in the discussion.

Results, page 4, line 105: primary outcome – please adjust also to primary site of infection

Results page 5, line 117 – please provide as supplemental data the univariable analysis that lead to the election of variables to include in the multiple regression model for each secondary outcome: free from invasive ventilation at 14 and 28 days and mortality rate at 14 days.

Methods, page 8, line 257-259 – did not understand the difference between 28 days’ mortality and discharged alive from the ICU at day 28, please explain

Author Response

Reviewer(s)' Comments to Author:

#1 Reviewer’s comments

comments 1) In introduction page 1-2, lines 44-47, the authors refer as a limitation to clinical implementation of continuous infusion of piperacillin-tazobactam concerns about drug stability, drug-drug incompatibility of antibiotics and concerns about the emergence of antimicrobial resistance. These are very important issues and the authors should return to this subject in the results (if they have any, namely practical problems concerning drug-drug incompatibility) or in the discussion.

Response: Issues with drug stability were not reported during the study period. We educated physicians and nurses in the medical ICUs about drug–drug compatibility before implementing the protocol and provided the relevant information table. Nevertheless, only 29.9% of 157 patients received continuous infusion of piperacillin-tazobactam during the study period, reflecting the insufficient application of continuous infusion in real practice. We have added sentences related to the clinical implementation of continuous infusion in the Materials and Methods (Page 9, Line 268-269), Results (Page 2, Line 58-59), and Discussion (Page 8, Line 226-228).

comments 2) Results, page 4, line 105: primary outcome – please adjust also to primary site of infection

Response: We performed the analysis, as you recommended, to adjust the primary outcome according to the primary site of infection. The results were similar to previous results and showed that continuous rather than intermittent infusion of piperacillin-tazobactam was significantly associated with a lower mortality rate at 28 days (adjusted HR 0.31; 95% CI: 0.12–0.78; p = 0.013). Therefore, we corrected the sentences in the Results (Page 4, Line 111 and Page 5, Line 115) and in Supplemental Table 1 (Supplemental Table 1).

comments 3) Results page 5, line 117 – please provide as supplemental data the univariable analysis that lead to the election of variables to include in the multiple regression model for each secondary outcome: free from invasive ventilation at 14 and 28 days and mortality rate at 14 days.

Response: We provided the supplementary data for the univariable and multivariable analysis of freedom from mechanical ventilation at 14 and 28 days, and discharge alive from the ICU at 14 and 28 days (Supplemental Table 2, 3, 4, and 5).

comments 4) Methods, page 8, line 257-259 – did not understand the difference between 28 days’ mortality and discharged alive from the ICU at day 28, please explain

Response: Mortality at 28 days and discharge alive from the ICU at 28 days are both time-to-event results. Outcome mortality at 28 days consists of survival until 28 days (time) and death (event), whereas discharge alive from the ICU at 28 days is a composite outcome combining mortality and ICU length of stay. The main event is a discharge from the ICU within 28 days, and the time of data collection is calculated as the number of days in the ICU during the first 28 days after inclusion. Any patient death within 28 days is considered a competing risk event with zero ICU days. To help readers understand, we added sentences about the secondary outcomes in the Materials and Methods (Page 9, Line 275-279).

Reviewer 2 Report

In this article, Hyun et al. performed a retrospective study in which they analyzed the effectiveness of continuous vs. intermittent infusion of piperacillin-tazobactam in patients with sepsis. A total of 157 patients (47 in the continuous and 110 in the intermittent groups) were evaluated. They observed that patients in the continuous group showed significantly lower mortality at 28 days than those in the intermittent group.

The study is well performed and comprehensive in terms of analysis of variables as well as the number of parameters considered. The study is certainly important as it has the potential to define a treatment paradigm for patients with sepsis. The authors need to address the following minor concerns to make their study robust.

1. The patients were also treated with other drugs, including teicoplanin, levofloxacin, and trimethoprim-sulfamethoxazole. The authors need to discuss the contribution/interference of these agents to their study outcomes in detail.

2. In line 69, the authors posit that “Laboratory variables associated with infection 69 were similar between the two groups”. It is unclear which variables the authors are talking about.

3. This study and many previous studies suggest that continuous infusion of piperacillin-tazobactam is beneficial. The authors need to discuss-

a) How is their study different from the previous reports?

b) Why do some patients still receive intermittent infusion despite lower efficiency?

4. For ease of understanding for the reader, the authors need to explain the following:

a) What do CRP, procalcitonin, and lactate levels indicate?

b) What are Sequential Organ Failure Assessment (SOFA) scores and Acute  Physiology and Chronic Health Evaluation â…¡ _(APACHE â…¡), and how are they performed?

5. There are minor grammatical errors throughout the article, including spelling, additional spaces and periods, tenses, etc. Thorough proofreading is warranted before final submission.

Author Response

#2 Reviewer’s comments

comments 1) The patients were also treated with other drugs, including teicoplanin, levofloxacin, and trimethoprim-sulfamethoxazole. The authors need to discuss the contribution/interference of these agents to their study outcomes in detail.

 Response: As you noted, most patients in this study simultaneously received other antibiotics, which may have affected the benefit on outcomes. However, the interference of other antibiotics with outcomes seemed to be insignificant. First, there was no difference in the proportion of other antibiotic use between the two groups. In addition, the use of other antibiotics was not significantly associated with mortality at 28 days, or discharge from the ICU at 14 and 28 days, in the multivariable analysis. Although there was an association between other antibiotic use and ventilator weaning, patients with continuous rather than intermittent infusion had a significantly higher likelihood of ventilator weaning at 14 days, even after adjustment for the use of other antibiotics. We have modified sentences about concomitant antibiotics in the limitations section of the Discussion (Page 8, Line 216-226).

comments 2) In line 69, the authors posit that “Laboratory variables associated with infection 69 were similar between the two groups”. It is unclear which variables the authors are talking about.

 Response: We have corrected the sentence to make the intention clear (Page 2, Line 70-71).

comments 3) This study and many previous studies suggest that continuous infusion of piperacillin-tazobactam is beneficial. The authors need to discuss-

  1. a) How is their study different from the previous reports?
  2. b) Why do some patients still receive intermittent infusion despite lower efficiency?

 Response a): This study had strengths that were different from previous studies. First, previous studies investigating the therapeutic effects of continuous beta-lactam antibiotic infusions on sepsis had heterogeneous populations or used various types of beta-lactam, whereas our study included only patients with sepsis (based on the Sepsis-3 definition) treated with piperacillin-tazobactam. Second, most previous reports did not analyze mortality as the primary outcome, whereas our study focused on improved survival in patients with sepsis. Also, previous studies did not show an advantage for continuous infusion on survival. We added these points about our study strengths to the Discussion (Page 8, Line 230-233).

Response b): As mentioned above, most previous studies that compared mortality between continuous and intermittent infusions of beta-lactam antibiotics as a secondary outcome did not show a survival benefit for continuous infusion because of multiple factors, such as heterogeneous patient groups, several types of beta-lactams, or small sample sizes. It was only recently that meta-analyses reported a superior effect on mortality for continuous versus intermittent infusion. In addition, several obstacles limit the wide application of continuous infusion in actual practice, as mentioned in the Introduction. In this study, only 30% of patients received continuous infusion. We now require multicenter, double-blinded, prospective, randomized studies to confirm the benefits of continuous infusion stated in the Conclusions (Page 10, Line 326-327).

comments 4) For ease of understanding for the reader, the authors need to explain the following:

  1. a) What do CRP, procalcitonin, and lactate levels indicate?
  2. b) What are Sequential Organ Failure Assessment (SOFA) scores and Acute Physiology and Chronic Health Evaluation â…¡_(APACHE â…¡), and how are they performed?

Response a): We have added a sentence about CRP, procalcitonin, and lactate levels in the Materials and Methods (Page 9, Line 280-282).

Response b): SOFA and APACHE II scores are commonly used as severity scoring systems in ICUs: a high score corresponds to an increased risk of death. We modified the sentence in the Materials and Methods to clarify the meaning of the two scoring systems (Page 9, Line 288), and we added the score-calculation formulas to the Supplementary Materials (Supplemental Table 6 and 7).

comments 5) There are minor grammatical errors throughout the article, including spelling, additional spaces and periods, tenses, etc. Thorough proofreading is warranted before final submission.

Response: To correct minor grammatical errors throughout the article, once again, this manuscript has been carefully reviewed by an experienced editor whose first language is English and who specializes in editing papers written by scientists whose native language is not English.

Reviewer 3 Report

In their manuscript “Continuous piperacillin-tazobactam infusion improves clinical outcomes in critically ill patients with sepsis: A retrospective, single-center study,” Hyun et al perform a retrospective study comparing outcomes in septic adult patients receiving intermittent piperacillin-tazobactam or continuous piperacillin-tazobactam. They report improved 28 day mortality in recipients of continuous infusion when adjusted for confounding factors. This is an interesting report suggesting continuous piperacillin-tazobactam may be associated with better outcomes in septic adults, though it is limited by the retrospective design. It does provide data supporting prospective, randomized control trials of intermittent versus continuous piperacillin-tazobactam.In their manuscript “Continuous piperacillin-tazobactam infusion improves clinical outcomes in critically ill patients with sepsis: A retrospective, single-center study,” Hyun et al perform a retrospective study comparing outcomes in septic adult patients receiving intermittent piperacillin-tazobactam or continuous piperacillin-tazobactam. They report improved 28 day mortality in recipients of continuous infusion when adjusted for confounding factors. This is an interesting report suggesting continuous piperacillin-tazobactam may be associated with better outcomes in septic adults, though it is limited by the retrospective design. It does provide data supporting prospective, randomized control trials of intermittent versus continuous piperacillin-tazobactamIn their manuscript “Continuous piperacillin-tazobactam infusion improves clinical outcomes in critically ill patients with sepsis: A retrospective, single-center study,” Hyun et al perform a retrospective study comparing outcomes in septic adult patients receiving intermittent piperacillin-tazobactam or continuous piperacillin-tazobactam. They report improved 28 day mortality in recipients of continuous infusion when adjusted for confounding factors. This is an interesting report suggesting continuous piperacillin-tazobactam may be associated with better outcomes in septic adults, though it is limited by the retrospective design. It does provide data supporting prospective, randomized control trials of intermittent versus continuous piperacillin-tazobactam

Author Response

#3 Reviewer’s comments

comments 1) In their manuscript “Continuous piperacillin-tazobactam infusion improves clinical outcomes in critically ill patients with sepsis: A retrospective, single-center study,” Hyun et al perform a retrospective study comparing outcomes in septic adult patients receiving intermittent piperacillin-tazobactam or continuous piperacillin-tazobactam. They report improved 28 day mortality in recipients of continuous infusion when adjusted for confounding factors. This is an interesting report suggesting continuous piperacillin-tazobactam may be associated with better outcomes in septic adults, though it is limited by the retrospective design. It does provide data supporting prospective, randomized control trials of intermittent versus continuous piperacillin-tazobactam.

Response: We thank you for your careful review. As you noted, we also proposed further multicenter, prospective, randomized studies in the Conclusions (Page 10, Line 326-327).